# Endometriosis-Associated Ovarian Cancer: From Molecular Pathologies to Clinical Relevance

**DOI:** 10.3390/ijms25084306

**Published:** 2024-04-13

**Authors:** Sophie Charlotte Steinbuch, Anne-Marie Lüß, Stephanie Eltrop, Martin Götte, Ludwig Kiesel

**Affiliations:** 1Department of Obstetrics and Gynecology, University Hospital Münster, Albert-Schweitzer-Campus 1, 48149 Münster, Germany; 2Cells-in-Motion Interfaculty Centre (CiMIC), University of Münster, 48149 Münster, Germany

**Keywords:** endometriosis, ovarian cancer, EAOC

## Abstract

Endometriosis is a chronic condition affecting reproductive-aged women, characterized by the growth of ectopic endometrial tissue. Despite being benign, endometriosis is associated with an increased risk of certain cancers, including endometriosis-associated ovarian cancer (EAOC). Ovarian cancer is rare, but more common in women with endometriosis, particularly endometrioid and clear-cell carcinomas. Factors such as hormonal imbalance, reproductive history, environmental exposures, and genetic predisposition contribute to the malignant transformation of endometriosis. Thus, understanding potential risk factors causing malignancy is crucial. Over the past few decades, various genetic mutations, microRNAs, as well as tumor microenvironmental factors have been identified, impacting pathways like PI3K/AKT/mTOR, DNA repair mechanisms, oxidative stress, and inflammation. Thus, this review aims to summarize molecular studies involved in EAOC pathogenesis as potential therapeutic targets. However, further research is needed to better understand the molecular and environmental factors driving EAOC development, to target the susceptibility of endometriotic lesions to malignant progression, and to identify effective therapeutic strategies.

## 1. Introduction

Endometriosis is a common, benign chronic disease in women of reproductive age, where endometrial tissue grows outside the uterine cavity in the entire abdominal and pelvic cavity. Since this ectopic tissue underlies the hormonal cycle with estrogen-induced proliferation, symptoms are associated with the menstrual cycle (dysmenorrhea, hypermenorrhea, menorrhagia) but also depend upon the localization (dysuria, dyspareunia, dyschezia). In addition to chronic pain, endometriosis is associated with infertility and an elevated risk for certain cancers including endometriosis-associated ovarian cancer (EAOC) [1,2].

### 1.1. Diagnosis

Generally, endometriosis is diagnosed based on typical signs and symptoms according to the European Society of Human Reproduction and Embryology (https://www.eshre.eu, accessed on 8 April 2024) [3], the American College of Obstetricians and Gynecologists (https://www.acog.org, accessed on 8 April 2024), or the Asian Society of Endometriosis and Adenomyosis (http://endometriosis-adenomyosis.asia, accessed on 8 April 2024).

Diagnostic measures include a clinical examination and imaging (ultrasound or magnet resonance imaging) but can be extended with a diagnostic laparoscopy to confirm the histology [3]. Diagnosing endometriosis with architectural atypia is crucial as it may indicate a precursor lesion of ovarian cancer, prompting pathologists to carefully examine surgical specimens to identify patients with hyperplasia-type endometriosis who could be at increased risk of developing endometriosis-associated ovarian cancer (EAOC) [4]. To distinguish between EAOC and benign endometriotic cysts or borderline tumors, the Sampson and Scott diagnostic criteria are used, including (1) the presence of endometriosis and cancer within the same ovary, (2) exhibiting a comparable histological structure to endometrial stroma, (3) while excluding the possibility of a metastatic tumor affecting the ovary, and (4) histopathological proofing of the progression from benign endometriosis to malignancy [5,6].

### 1.2. Risk Factors

Overall, the risk for developing ovarian cancer is low, with a cumulative risk of under 1% [7]. However, women with endometriosis are more likely to develop ovarian cancer [8], especially the endometrioid and clear-cell histotype [9,10]. Hereby, ovarian clear cell carcinomas (OCCCs) are per-definition high-grade ovarian carcinomas with a poor prognosis in advanced stages, whereas endometrioid ovarian carcinomas (EnOCs) can manifest as either low- or high-grade tumors [11]. However, there is no difference in the histological profiles of EAOC and ovarian cancer arising in endometriosis, indicating a comparable etiopathological mechanism [12].

Certain risk factors like a larger number of ovulatory cycles (early menarche/late menopause) [13], less pregnancies (infertility, nulliparity) [14], and a shorter breastfeeding duration [15] contribute to the malignant transformation of endometriosis into ovarian cancer [16]. Generally, a hormonal imbalance with an estrogen excess and progesterone deficit promotes the development of EOAC [17]. Thus, hormonal and reproductive factors such as the use of oral contraceptives [18,19], tubal ligation [20], polycystic ovarian syndrome (reviewed by Throwba et al. [21]), recreational physical activity [22], hormone therapy [23], and obesity [24,25,26] affect the risk for developing endometriosis and ovarian cancer.

Furthermore, endometrial cysts are a significant risk factor for developing cancer and thus should be carefully monitored [27]. However, the sonographic differentiation between benign endometrioma (“chocolate cysts”) and early-stage ovarian cancer remains challenging [11].

Besides hyperestrogenism and endometrial cysts, oxidative stress and inflammation are discussed as contributors to malignant transformation [17,28]. Therefore, other risk factors include exposure to environmental toxins, including dioxins [29,30], phthalates [31], alcohol [32] and coffee consumption [33], and smoking [34], as well as improper nutritional habits [35,36]. Finally, genetic predisposition has a large influence on the transition from benign endometriosis to malign ovarian cancer.

## 2. Molecular Pathologies

Almost one hundred years ago, John A. Sampson first described ectopic endometrium-like tissue as the cause of ovarian carcinoma [5] and concluded that the “Metastatic or Embolic Endometriosis, [is] due to the Menstrual Dissemination of Endometrial Tissue into the Venous Circulation” [37]. During the subsequent century, this concept of retrograde menstruation [38], followed by the implantation of desquamated endometrial cells in the peritoneal cavity and the sequential transformation of those endometriotic lesions in the ovarian endometrium into ovarian cancer through atypical endometriosis was further examined [39]. However, only 2% of women with laparoscopically confirmed endometriosis develop ovarian cancer [40]. While the exact etiopathology is still unknown, both endometrioid tissue intrinsic and microenvironmental factors are discussed as causal factors for survival in the peritoneum and malignant transformation [39].

For instance, the increased occurrence of chromosomal abnormalities observed in ovarian endometriosis compared to extragonadal endometriosis implies that the ovarian stromal environment may influence the initiation of genetic alterations, potentially progressing to invasive cancer [41].

### 2.1. Genetic Mutations

Hereby, several genetic mutations were identified as key players in the malignant conversion of endometriosis and manifestation of EAOC (Figure 1). In general, it is widely known that p53 and K-ras mutations and microsatellite instability promote tumor induction and growth, which is also the case for the malignant transformation of endometriosis [42,43,44]. Further common driver mutations in endometriosis include ARID1A, PIK3CA, and PPP2R1A [45,46]. However, BRCA mutations, commonly observed in ovarian carcinomas, are predominantly associated with serous ovarian cancer and are thus less frequent in EAOC [45,47,48].

#### 2.1.1. Chromatin Remodeling

Comparing shared genetics in uterine endometria, endometriosis, and EAOC, almost half of clear-cell carcinomas and endometrioid carcinomas harbored mutations in the tumor-suppressor gene ARID1A (AT-rich interactive domain-containing protein 1A) [46], encoding BAF250a—a main component of the ATP-dependent chromatin remodeling complex SWI/SNF involved in the transcriptional activation of chromatin-repressed genes [49]. The loss of BAF250a in tissues of patients with EAOC was associated with the elevated expression of the phosphorylated H2A histone family member X γH2AX (S139), involved in chromatin remodeling and the repair of DNA double-strand breaks [50], as well as a higher expression of the pro-apoptotic regulators BIM (Bcl-2 Interacting Mediator of cell death) and BAX (Bcl-2-associated X protein), but lower expression of the anti-apoptotic gene Bcl-2 [51]. These findings indicate an initiation of DNA damage response and apoptosis pathways, potentially as a response to genomic instability at an early stage in precancerous lesions [51]. Furthermore, ARIDA1A shares some common downstream targets with p53, including CDKN1A and SMAD3 [52]. Thus, the loss of ARIDA1A function leads to the dysregulation of p53-controlled genes [52].

#### 2.1.2. PI3K/AKT/mTOR Pathway

Among the subset of clear-cell EAOC, somatic mutations of the PIK3CA gene, encoding a catalytic subunit of phosphatidylinositol-3 kinases (PI3K), occur often as an early event [53,54], frequently coexisting with the loss of ARID1A protein expression [55], and may have syngeneic effects [56]. Other early signatures found in ARID1A-deficient carcinomas include the activation of the RAC (Rho family)-alpha serine/threonine-protein kinase via the increased expression of AKT1 and phosphorylation (pAKT) [49,51]. Furthermore, the differential expression of components involved in the mechanistic target of rapamycin (mTOR) pathway seem to connect endometriosis and the development of ovarian cancer [57].

A less common mutation in EOAC and ovarian clear-cell carcinoma (OCCC), detected in approximately 16–19% of OCCC cases [58,59], affects the function of the oncogene PPP2R1A, encoding a regulatory subunit of serine/threonine phosphatase 2 (PP2A) [60], which is a negative regulator of cell growth [61].

Altogether, the PI3K/AKT/mTOR pathway is a major regulator of the cell cycle; thus, alterations in gene regulation due to mutations contribute to the development and progression of ovarian cancer (reviewed by Aziz et al. [62]) as well as in the transformation process from a healthy endometrium to endometriosis and EAOC (reviewed by Driva et al. [63]). In contrast, the activity of the phosphatase and tensin homolog (PTEN), the counterpart of the PI3K/AKT pathway, was reduced due to PTEN silencing in EAOC, thus alleviating the negative control of PTEN on cell growth and division [64,65,66].

#### 2.1.3. Other Genetic Alterations

Overall, the development of endometriosis-associated ovarian cancer is associated with genetic aberrations in multiple pathways, promoting malignant transformation and cancer cell growth. Hereby, the genetic profile differs between benign ovaries and ovarian endometriosis from EAOC and ovarian cancer [67]. Er et al. identified several more mutated genes belonging to the Wnt pathway, the mitogen-activated protein kinase (MAPK)/extracellular signal-regulated kinase (ERK) pathway, the Notch signaling pathway, cell cycle control, and the mismatch repair system via targeted next-generation sequencing [68]. Of note, the Notch signaling pathway is also dysregulated in endometriosis and has been mechanistically linked to its pathogenesis [69,70]. Other sequencing approaches highlighted the genomic heterogeneity and the diversity of cancer-associated mutations in ovarian endometriotic and normal uterine endometrial epithelium samples from patients [71]. Hereby, shared mutations among cases of EAOC displayed higher median variant allele frequencies, suggesting positive selection for clones with these mutations [71]. Therefore, the future whole-genome sequencing of EAOCs will certainly detect more differential expressed genes, whose relevance needs to be confirmed in consecutive experimental studies.

### 2.2. Epigenetic Reprogramming

In addition to genetic mutations, epigenetic mechanisms are involved in the malignant transformation of endometriosis to EAOC. For instance, the transcriptional inactivation of the MLH1 gene, encoding a DNA mismatch repair (MMR) protein, by promoter hypermethylation was reported [72], leading to microsatellite instability and the accumulation of spontaneous mutations, thus promoting progression towards EAOC. Further differentially methylated genes were identified as potential candidates in facilitating the malignant transformation, including the RASSF2 gene [73], encoding the KRAS-specific effector protein Ras association domain-containing protein 2, as well as the RUNX3 gene [74], which encodes the tumor suppressing Runt-related transcription factor 3.

### 2.3. The Tumor Microenvironment

#### 2.3.1. Estrogen Concentration

In addition to tumor-intrinsic genetic factors, the tumor microenvironment is an emerging hallmark of cancer [75] and thus shapes EAOCs. One major player in the malignant transformation to EAOC is the estrogen concentration in the surrounding milieu, either exogenously provided by estrogen replacement therapy [76,77] or endogenously produced by the ovaries [78]. Hereby, high estrogen levels promote the proliferation of endometriotic cells via binding to the non-classical, G-protein-coupled estrogen receptor 1, followed by the activation of various signaling pathways such as the PI3K/AKT/mTOR [79], as reviewed by Kozieł et Piastowska-Ciesielska [80]. Furthermore, an imbalance between the expression levels of the classical estrogen receptors ER-alpha and ER-beta, favoring ER-alpha dominance due to the overexpression of ER-alpha [81] or loss of the anti-cancerous ER-beta [82,83,84], contributes to (ovarian) carcinogenesis [85]. While supporting the role of ER-beta as a tumor suppressor via the inhibition of ER-alpha-induced cell proliferation [86], some researchers assume a contrary shift from ER-alpha to ER-beta signaling during the estrogen-dependent progression of endometriosis to EAOC [87,88]. Altogether, the role of estrogen signaling in EAOC is more complex, especially considering the fact that the signaling is affected by the nutritional status [89], oxidative stress [90], and surrounding cells [91], but that estrogen vice versa shapes the cellular metabolism [92], the epithelial to mesenchymal transition [93], angiogenesis [94], and viability and invasiveness [95].

Further, estrogen influences epigenetic processes by regulating the DNA methyltransferase 1 (DNMT1)-dependent hypermethylation of the runt-related transcription factor 3 (RUNX3) in the malignant transformation of ovarian endometriosis [96].

#### 2.3.2. microRNAs

Besides estrogen, microRNAs (miRNAs) are other important, post-transcriptional factors regulating the gene expression and thus are considered as “novel biomarkers” in endometriosis and EAOCs [97]. These small non-coding RNA molecules, typically consisting of 21 to 25 nucleotides, bind to complementary sequences in the messenger RNA and thus lead to RNA silencing by mRNA degradation or translational repression [98,99,100,101]. Over the past few decades, the complex dysregulation of microRNA expression has been observed in several human cancers [102], including differential expressed miRNAs in ovarian cancer [103,104], endometriosis [105,106], and EAOC [97,107].

In ovarian cancer, the miR-200 and lethal-7 families were the groups with the most significantly altered expression [108]. In this context, the let-7 family members were identified as cell cycle regulators as well as suppressors of oncogenes like the high-mobility group AT-hook 2 (HMGA-2), K-Ras, and Myc [108,109]. miRNA-200 is involved in inhibiting the epithelial-to-mesenchymal transition [110], while its downregulation induces tumor progression [111,112]. Notably, miR-200b also has a role in the pathogenesis of endometriosis, as it regulates the stem cell phenotype, proliferation, invasiveness, and growth of invasive protrusions of endometriotic cells by targeting ZEB1, ZEB2, and KLF4 [113,114].

Interestingly, Szubert et al. examined a decrease in the expression of miR-31-3p and miR-200b in cancerous lesions in contrast to normal ovarian tissue and endometriosis tissue [115]. miR-31 inhibits the tumor suppressor ARID1A, thus elevating the oncogenicity and stemness of head and neck squamous cell carcinoma [116]. Furthermore, miR-31 activates hypoxia-inducible factor under normoxic conditions by targeting the 3′untranslated region of the regulator FIH (factor-inhibiting hypoxia-inducible factor), leading to increased vascular endothelial growth factor (VEGF) production [117]. In turn, VEGF overexpression is associated with endometriosis [118] and the progression to EAOC [119]. Moreover, decreased levels of various other microRNAs such as miR-17-5p, miR-20a, miR-222, and miR-125a have been associated with angiogenesis in endometriosis through the regulation of runt-related transcription factor 1 (RUNX1), connective tissue growth factor (CTGF), thrombospondin-1 (TSP-1), or vascular endothelial growth factor-A (VEGF-A) expression [106,120,121].

#### 2.3.3. Oxidative Stress

Recently, Marí-Alexandre et al. addressed the complex interaction between microRNAs and oxidative stress in a review, shedding light on their interplay during endometriosis, EAOC, and high-grade serous ovarian cancer (HGSOC) [39]. In short, microRNAs regulate the expression of reactive oxygen species (ROS)-producing or detoxifying enzymes, whereas oxidative stress induces the production of responsive microRNA via ROS-sensitive transcription factors [39]. However, oxidative stress itself also affects the gene expression of EAOC-related genes directly. For example, ARD1A expression is downregulated via oxidative stress, characterized by the enzymatic activity of manganese superoxide dismutase and by the formation of malondialdehyde due to ROS-dependent lipid degradation or peroxidation [122]. Mandai et al. even propose that persistent oxidative stress in endometriotic cysts is a causal contributor to their carcinogenic transformation [123]. A probable reason for these elevated ROS levels in the intra-cystic fluid within the ovarian endometrioma could result from the release of free iron during monthly bleeding via the Fenton reaction [39,124,125].

#### 2.3.4. Inflammation

In addition to oxidative stress, inflammation promotes the carcinogenesis of EAOCs [123] by creating a pro-tumorigenic microenvironment characterized by DNA damage, tissue remodeling, immune suppression, angiogenesis, and epithelial–mesenchymal transition. In this context, several inflammatory cytokines [67], complement factors [126], inflammasome-related genes [127], as well as distinct immunologic and inflammatory signatures [91] were identified, indicating the relevance of the immune system in endometriotic cells and EAOCs. Recently, Linder et al. pointed out that genes related to the immune system are commonly mutated yet conserved between ovarian endometriosis and the subsequent development of ovarian carcinoma, suggesting that genetic alterations impacting immune response occur early and play a crucial role in these conditions [128].

#### 2.3.5. Nutrient Availability

An emerging hallmark of cancer involves the tumor’s adaptation to the local nutrient availability through metabolic reprogramming [75,129]. On the one hand, endometriotic cells prefer aerobic glycolysis to generate energy, even in the presence of oxygen (called the “Warburg effect”) [130], facilitating their survival in extrauterine sites under oxidative and hypoxic stress [131]. On the other hand, recent studies demonstrated that oxidative phosphorylation (OXPHOS) is altered in ovarian cancer [132,133,134]. Overall, Dar et al. (2017) assume that ovarian cancer cells exhibit heterogeneity in energy metabolism by utilizing both glycolysis and OXPHOS to enhance their “cellular fitness” and chemoresistance as part of their survival strategy [135].

In general, cancer cell proliferation requires not only energy (ATP), but also the conversion of available nutrients into biomass [136]. Therefore, metabolite flexibility allows cancer cells to adapt to their environment [137], characterized by site-specific levels of glucose, lipids, and amino acids. However, subtypes of ovarian cancer display different metabolic preferences (as reviewed, for instance, in [133,138,139]), with no specific data available for EAOC.

### 2.4. The Complex and Heterogenous Nature of Cancer

To summarize, EAOC exemplifies the intricate and diverse landscape of cancer, marked by intra-tumoral, inter-tumoral, and inter-patient heterogeneity [128]. Within tumors, diverse cell populations coexist, exhibiting distinct genetic mutations, epigenetic alterations, and phenotypic characteristics. Inter-tumoral variability extends to differences between individual tumors, contributing to varied clinical presentations and responses to therapy. Moreover, the inter-patient tumor heterogeneity highlights the unique molecular signatures and disease trajectories observed among patients with EAOC. To conclude, this multifaceted nature of EAOC mirrors the complexity of understanding and effectively treating ovarian cancer, necessitating personalized approaches tailored to individual tumor characteristics. In particular, metastatic involvement in EAOC represents a complex clinical scenario associated with poor outcomes [140], thus requiring a multidisciplinary approach and targeted treatment strategies.

## 3. Therapeutic Strategies

### 3.1. Endometriosis

As a primary treatment, surgical removal is very common [11]—often through a laparoscopic approach—involving the ovaries (oophorectomy), fallopian tubes (salpingectomy), uterus (hysterectomy), and the surrounding tissue, as well as lymph nodes (lymphadenectomy). In patients with endometriosis-associated pain, surgical treatment combined with analgesics or hormone treatment (including combined hormonal contraceptives, progestogens, gonadotropin-releasing hormone (GnRH) agonist/antagonists, or aromatase inhibitors) is recommended (https://www.eshre.eu/Guideline/Endometriosis, accessed on 8 April 2024). Nevertheless, information is still lacking about the advantage of surgery (“unilateral salpingo-oophorectomy or cystectomy/partial ovarian excision”) compared to conservative surveillance on mortality from EAOC in patients with (asymptomatic) endometriosis [141].

### 3.2. Prevention of Cancer in Patients with Endometriosis

Younis states that, even if the likelihood of developing EAOC is nearly doubled in women with endometrioma, the overall lifetime risk remains minimal and that EAOC occurs more frequently in older women, with around one-third of EAOC cases diagnosed in the premenopausal stage (below 50 years), 2.10% cases in women under the age of 45, and only 0.017% of all cases in women below 40 years [142]. Thus, Younis suggests conservative management with a focus on transvaginal ultrasound to differentiate between benign, “homogenous cystic ‘ground glass’”-appearing endometrioma and EAOC [142]. Only EAOC-suspicious findings, usually presenting as a large, vascularized, papillose, unilateral cysts (>9 cm) with solid features, should be further characterized via T2-weighted magnetic resonance imaging [142,143].

Likewise, the ESHRE guideline concludes that “both diagnostic laparoscopy and imaging combined with empirical treatment (hormonal contraceptives or progestogens) can be considered in women suspected of endometriosis” [3]. Furthermore, the guideline development group highlights the importance of an individual case decision regarding long-term monitoring and empirical treatment based on specific risk factors [3,144]. In conclusion, the decision for the complete excision of all visible endometriotic lesions as a potential benefit to reduce the risk for developing EAOC should always be weighed against the risks of surgery like pain, morbidity, and the ovarian reserve [3]. Preserving the ovarian reserve is especially important in younger women due to reproductive benefits [145], but also due to the increased risk of coronary heart disease associated with premature hypoestrogenism [141,146]. Furthermore, hypoestrogenism leads to decreased bone mineral density, resulting in osteopenia and osteoporosis and subsequently an increased risk of pathological fractures [147].

Therefore, the use of combined oral contraceptives is discussed as an alternative early intervention in adolescents suffering from acute dysmenorrhea episodes—a potential sensitizer to chronic pelvic pain as well as an indicator for early-onset endometriosis [148]. In addition to alleviating the acute symptoms, reducing the amount of menstrual bleeding [149], and preventing the development of endometriosis [148], the long-term use of combined hormonal contraceptives decreases the risk of certain cancers, including ovarian cancer [150,151,152,153]. Overall, the long-term use of oral contraceptives might be a suitable preventor of EAOC by limiting disease progression while preserving future reproductive potential [148].

### 3.3. Treatment of Diagnosed Endometriosis-Associated Ovarian Cancer

A completely different therapeutic situation occurs in patients with confirmed endometriosis-associated ovarian carcinoma (Figure 2). Here, the therapy typically involves a multidisciplinary approach including surgery, chemotherapy, radiation therapy, targeted therapy, and/or hormonal therapy. However, the individual treatment plan depends on various factors such as the stage of cancer, the extent of spread, the patient’s overall health, and their fertility desires. Therefore, the therapy of patients with EAOC and other ovarian cancers differs from their histological subtype [154], namely, the high-grade serous ovarian carcinoma (HGSOC), low-grade serous carcinoma of the ovary (LGSOC), clear-cell ovarian carcinoma (CCOC), endometrioid ovarian carcinoma, mucinous carcinoma, and borderline ovarian tumors, as well as germ cell tumors and stromal tumors [155]. Nevertheless, primary ovarian cancer and EAOC with the same pathological type share a similar origin [12,156], and thus may result in comparable treatment strategies.

#### 3.3.1. Surgical Treatment

Primary treatment for epithelial ovarian carcinoma and endometriosis-associated ovarian carcinoma typically involves initial debulking surgery followed by platinum-based chemotherapy [157]. Cytoreductive therapy in ovarian cancer usually aims for the complete resection of tumor masses, while the surgical management of EAOC may also involve removing endometriotic lesions along with any associated ovarian masses. Given the younger age of patients with EAOC, it is essential to prioritize preserving ovarian function whenever feasible, particularly in women who desire to have children.

In general, ovarian debulking surgery in endometriosis-associated ovarian cancer has a greater success rate than in non-EAOC patients [158]. However, a stage-matched analysis revealed no difference in the effectiveness of the standard cytoreductive surgery followed by taxane- and platinum-based chemotherapy in patients with EAOC and non-EAOC [8]. Therefore, the better prognosis of EAOC is more likely due to the early-stage, low-grade disease [8,159,160].

#### 3.3.2. Chemotherapy

Besides surgery, adjuvant chemotherapy is a standard treatment option for ovarian cancer [161,162,163,164], including EAOC, particularly in advanced stages. Chemotherapeutic drugs commonly used in the treatment of ovarian cancer include platinum-based drugs like cisplatin and carboplatin, as well as taxanes such as paclitaxel [165,166,167,168,169]. However, the chemotherapy resistance of the ovarian clear-cell carcinoma subtype, also within EAOC, remains challenging and requires new therapeutic approaches.

#### 3.3.3. Hormonal Therapies

Hormonotherapy is another adjuvant systemic treatment option in endometrioid ovarian cancer exhibiting hormonal receptor positivity [155,163]. Thus, the hormone-receptor status has an impact on the treatment [170], as is known for other gynecological tumors. For instance, high expression levels of progesterone receptor in endometrioid ovarian carcinoma were associated with a favorable outcome and thus could be potential targets for endocrine therapy [171]. On the contrary, the loss of estrogen receptor alpha immunoreactivity [172,173] or the high expression of estrogen receptor beta and gamma [174] decreased overall survival in ovarian cancer. Therefore, the tumor expression of the progesterone receptor and estrogen receptor are prognostic biomarkers in certain ovarian cancer subtypes [175] but could also have some predictive value as biomarkers predicting the response to endocrine therapies [176]. Several biomarker studies have been conducted in the past few years, using endocrine agents such as letrozole, tamoxifen, aromatase inhibitors, and fulvestrant in ovarian cancer, as reviewed by Langdon et al. [176]. In addition, estradiol–triazole analogs were developed using a molecular modeling approach to target several proteins in the epidermal growth factor receptor/mitogen-activated protein kinase pathway synergistically in ovarian cancer [177].

#### 3.3.4. Targeted Therapies

To identify new therapeutic targets interfering with the underlying molecular pathology, several animal models were developed, including mice with a conditional doxycycline-induced Arid1a and/or Pten knockout [178], mice with endometrial autoimplantation and 7,12-Dimethylbenz[a]anthracene (DMBA) tumor induction [179].

In recent years, a functional precision oncology strategy was applied to identify patient specific drug regimens [154]. In this context, vascular endothelial growth factor (VEGF)/VEGFR pathway inhibitors such as bevacizumab [180,181,182] and poly(ADP-ribose) polymerase (PARP) inhibitors (like olaparib, niraparib, and rucaparib) are approved by the US Food and Drug Administration [183,184,185] and increasingly utilized in clinical settings for the treatment of ovarian cancer patients [186]. These anti-VEGF drugs interfere with tumor angiogenesis and normalize tumor vasculature [187], thus reducing tumor progression and enhancing drug susceptibility. Therefore, combining bevacizumab either with chemotherapy [180,181] or with PARP inhibitors [188,189,190] leads to synergistic effects. Interestingly, the differential expression of some microRNAs correlates to therapy response in ovarian cancer [191,192].

Especially patients with BRCA1/2-mutated tumors benefit from targeted therapies with PARP inhibitors due to the concept of synthetic lethality [193,194]: Tumor cells with a BRCA mutation lack the ability to repair DNA double-strand breaks via homologous recombination and are thus dependent on repairing DNA damage via the single-strand break (SSB) repair pathway [195]. By inhibiting the ribozyme PARP, mainly involved in SSB repair, tumor cells with BRCA mutations cannot repair DNA damage, leading to the accumulation of DNA damage and subsequently cell death without affecting normal cells [195].

To conclude, targeted therapies like PARP inhibitors are promising first-line treatment options in patients with BRCA-mutated ovarian cancer. However, further research and clinical trials in patients with EAOC are necessary to confirm the benefit of PARP inhibitors and bevacizumab in this clinical subgroup. In addition, targeted therapies using tyrosine-kinase inhibitors, MEK inhibitors, and PI3K/mTOR/Akt inhibitors are currently being tested in several ovarian cancer subtypes [155]. For example, the efficiency of the mTOR/AKT inhibitor temsirolimus in decreasing endometriotic cell proliferation was confirmed in a mouse model of deep-infiltrating endometriosis [196]. However, early clinical trials have shown that the efficacy of temsirolimus in treating ovarian cancer is limited [197], with no data available for patients with EAOC.

#### 3.3.5. Immunotherapy

Immunotherapy is emerging as a promising approach in ovarian cancer treatment, with its role continually evolving. Checkpoint inhibitors targeting the programmed cell death protein-1 (PD-1)/programmed death-ligand 1 (PD-L1) expression in EAOC are promising, as endometriosis-associated ovarian cancer expresses higher levels of PD-1/PD-L1 compared to benign lesions [198]. Additionally, the tumor mutation burden might predict the response to immunotherapy in EAOC [199], making it a potential biomarker for therapy susceptibility, as demonstrated in other tumors [200,201].

Using in vitro and in vivo models, Hsieh et al. revealed that the ARID1A-induced expression of HDAC6, resulting in IL-10 release and the subsequent M2 polarization of macrophages, is a new therapeutic target addressed by the HDAC inhibitor vorinostat [202].

## 4. Conclusions

In conclusion, the understanding of the molecular pathways and genetic aberrations involved in the development of endometriosis-associated ovarian cancer (EAOC) has significantly advanced, shedding light on potential therapeutic targets. Furthermore, the emerging role of the tumor microenvironment, estrogen signaling, oxidative stress, and inflammation highlights the complexity of EAOC pathogenesis, paving the way for novel therapeutic strategies including targeted therapies and immunotherapies.

In summary, while there may be similarities in the treatment approach for ovarian cancer and endometriosis-associated ovarian cancer, there are also important differences in terms of surgical considerations, chemotherapy regimens, and the potential role of hormonal therapy, reflecting the distinct etiology and characteristics of these diseases.

## Figures and Tables

**Figure 1 ijms-25-04306-f001:**
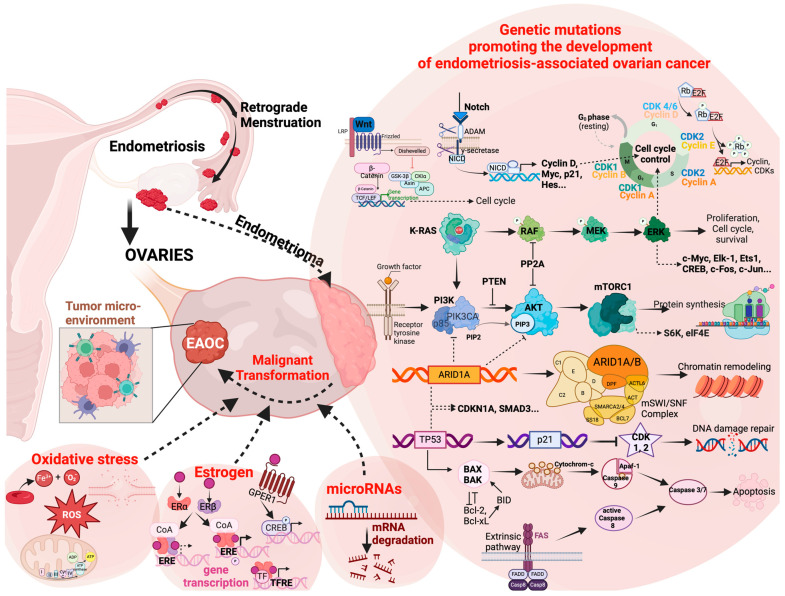
Molecular factors involved in the malignant transformation of endometrioma to endometriosis-associated ovarian cancer (EAOC). Through retrograde menstruation, endometrial tissue grows outside the uterus in ectopic locations including the ovaries. Hereby, several genetic alterations, microRNAs, local estrogen levels, and oxidative stress contribute to the malignant transformation of benign endometrial lesions to tumor progression. Common genetic mutations accumulating during the growth process affect the function of the proto-oncogene K-Ras (Kirsten rat sarcoma virus), the tumor suppressor gene P53 (TP53), the catalytic subunit alpha (PIK3CA) of the phosphatidylinositol 3-kinase (PI3K), as well as negative regulators of the PI3K/AKT/mTOR pathway, including the phosphatase and tensin homolog (PTEN) and a regulatory subunit of protein phosphatase 2 (PP2A) (encoded by the PPP2R1A gene). Furthermore, the AT-rich interactive domain-containing protein 1A (ARID1A) is frequently mutated, leading to the altered function of the mammalian SWItch/Sucrose Non-Fermentable (mSWI/SNF) complex, a subfamily of ATP-dependent chromatin remodeling complexes. Altogether, those mutated genes are mainly responsible for DNA damage repair, the regulation of apoptosis, protein synthesis, cell cycle control, proliferation, survival, and growth. Therefore, other signaling pathways such as the Wnt pathway (e.g., Wnt protein, lipoprotein receptor-related protein (LRP), T-cell factor/lymphoid enhancer factor, beta-catenin, but also its destruction complex [axin, adenomatosis polyposis coli (APC), glycogen synthase kinase 3 (GSK3) and casein kinase 1α (CK1α)]) and Notch pathways (neurogenic locus notch homolog protein-induced liberation of N-formylmaleamic acid amidohydrolase) may also be related to the development of EAOC. Alongside tumor-intrinsic genetic factors, microRNAs (miRNAs) serve as crucial post-transcriptional regulators of gene expression, exhibiting complex dysregulation in various human cancers, with the differential expression of specific families like miR-200, let-7, and miR-31 influencing epithelial-to-mesenchymal transition, oncogenicity, and angiogenesis. Furthermore, estrogen promotes endometriotic cell proliferation through various signaling pathways, while an imbalance favoring estrogen receptor (ER)-alpha dominance or loss of ER-beta contributes to carcinogenesis via an estrogen-responsive element (ERE)- or transcription factor (TF) response element (TFRE)-mediated gene transcription. In addition, estradiol binds to the G protein-coupled estrogen receptor 1 (GPER1 or GPR30), leading to the activation of the cAMP/PKA (protein kinase A)/CREB (cAMP response element-binding protein) pathway. Finally, oxidative stress plays a pivotal role in shaping EAOC by inducing DNA damage, inflammation, and aberrant cellular signaling and disrupting antioxidant defenses. [Figure created with BioRender].

**Figure 2 ijms-25-04306-f002:**
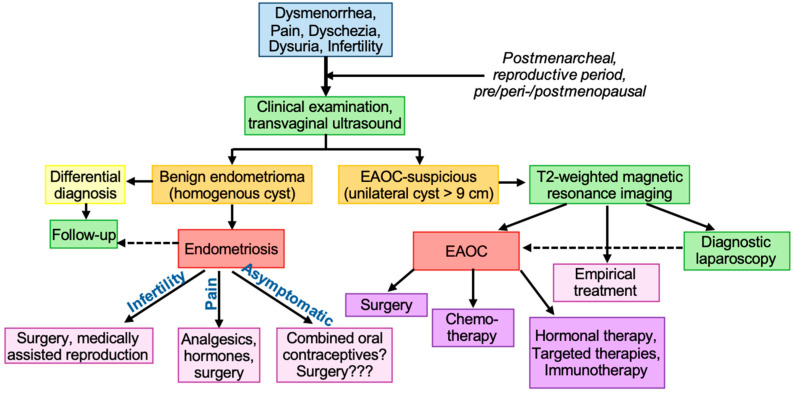
Diagnostic and therapeutic algorithm for endometriosis and endometriosis-associated ovarian cancer (EAOC). For further detailed information, see ESHRE guidelines.

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
