# Peer review of "Endometriosis-Associated Ovarian Cancer: From Molecular Pathologies to Clinical Relevance"

_ijms, 2024, doi:10.3390/ijms25084306_

Round 1

Reviewer 1 Report

Comments and Suggestions for Authors

While the review article offers a comprehensive retrospective analysis of Endometriosis-Associated Ovarian Carcinoma (EAOC), encompassing its pathogenesis and therapeutic modalities, several recommendations can be proposed to enhance the reader's comprehension of EAOC:

  1. 1. Given the well-established significance of BRCA1 and BRCA2 mutations as risk factors for ovarian cancer, it is advisable to include elucidative descriptions regarding the associations between BRCA mutations and EAOC within the section dedicated to genetic mutations.

  2. 2. As vascular endothelial growth factor (VEGF)/VEGFR pathway inhibitors such as Avastin and poly(ADP-ribose) polymerase (PARP) inhibitors are increasingly utilized in clinical settings for the treatment of ovarian cancer patients, augmenting the article with more detailed expositions concerning these therapeutic agents would be beneficial.

Author Response

Dear Reviewer,

Thank you for your insightful suggestions regarding our article.

We agree that providing elucidative descriptions of the associations between BRCA mutations and ovarian cancer is essential. While BRCA mutations are commonly observed in ovarian carcinomas, they are predominantly associated with serous ovarian cancer and are thus less frequent in EAOC. This distinction is crucial for a comprehensive understanding of the genetic landscape of ovarian cancer subtypes.

Regarding your second suggestion, we acknowledge the increasing utilization of vascular endothelial growth factor (VEGF)/VEGFR pathway inhibitors and poly(ADP-ribose) polymerase (PARP) inhibitors in clinical settings for the treatment of ovarian cancer patients. These therapeutic agents hold significant promise in improving patient outcomes by targeting specific molecular pathways involved in tumor progression and angiogenesis. We agree that augmenting the article with more detailed expositions concerning these therapeutic agents would be beneficial, particularly in elucidating their mechanisms of action, clinical efficacy, and potential synergistic effects when used in combination therapies.

In conclusion, we appreciate your suggestions and will incorporate detailed expositions regarding BRCA mutations and therapeutic agents such as PARP inhibitors and bevacizumab to provide a more comprehensive understanding of their roles in the treatment of ovarian cancer, including EAOC.

Thank you for your valuable input.

Yours sincerely,

Sophie Steinbuch

Reviewer 2 Report

Comments and Suggestions for Authors

Steinbuch et al. address the molecular and clinical alterations in Endometriosis-Associated Ovarian Cancer. However, it is important to understand the complex and heterogenous nature of cancer including ovarian cancer.

There are suggestions that can improve the impact of this paper.

1. A generalized section of intra-tumoral, inter-tumoral and inter-patient tumor heterogeneity can be added and make link with ovarian cancer.

2. A perspective on the influence of surgery and heterogeneity of Endometriosis-Associated Ovarian Cancer can be discussed.

3. A section on metabolic reprogramming and Endometriosis-Associated Ovarian Cancer can be discussed to understand molecular pathologies.

4. A section on epigenetic reprogramming and Endometriosis-Associated Ovarian Cancer can be presented.

5. A perspective on metastatic/secondary in Endometriosis-Associated Ovarian Cancer can be highlighted. 

Comments on the Quality of English Language

Moderate editing

Author Response

Dear Reviewer,

Thank you for your comments and suggestions regarding our manuscript on EAOC, emphasizing the importance of understanding its complex and heterogeneous nature. We appreciate your feedback, and added sections addressing the tumor heterogeneity, the metabolic and epigenetic reprogramming. Unfortunately, there are less data available for cancer metabolism in EAOC, so we highlighted the complexity of ovarian cancer and the different metabolic preferences of different subtypes in general. Furthermore, we emphasized the need for a multidisciplinary approach and personalized treatment strategies regarding metastatic EAOC.

In summary, we appreciate your feedback and are dedicated to incorporating these suggestions into our manuscript to enhance its comprehensiveness and relevance.

Thank you for your valuable input, which will undoubtedly strengthen the impact of our paper.

Sincerely,

Sophie Steinbuch